# Reproductive factors and the risk of incident dementia: A cohort study of UK Biobank participants

Jessica Gong[1]*, Katie Harris[1], Sanne A. E. Peters[1,2,3], Mark Woodward[1,2]

**1** The George Institute for Global Health, University of New South Wales, Newtown, New South Wales, Australia, **2** The George Institute for Global Health, Imperial College London, London, United Kingdom, **3** Julius Center for Health Sciences and Primary Care, University Medical Center, Utrecht University, Utrecht, the Netherlands

\* jgong@georgeinstitute.org.au

## Abstract

### Background

Women's reproductive factors have been associated with the risk of dementia; however, these findings remain uncertain. This study aimed to examine the risk of incident all-cause dementia associated with reproductive factors in women and the number of children in both sexes and whether the associations vary by age, socioeconomic status (SES), smoking status, and body mass index (BMI) in the UK Biobank.

### Methods and findings

A total of 273,240 women and 228,957 men without prevalent dementia from the UK Biobank were included in the analyses. Cox proportional hazard regressions estimated hazard ratios (HRs) for reproductive factors with incident all-cause dementia. Multiple adjusted models included age at study entry, SES, ethnicity, smoking status, systolic blood pressure, BMI, history of diabetes mellitus, total cholesterol, antihypertensive drugs, and lipid-lowering drugs. Over a median of 11.8 years follow-up, 1,866 dementia cases were recorded in women and 2,202 in men. Multiple adjusted HRs ((95% confidence intervals (CIs)), $p$-value) for dementia were 1.20 (1.08, 1.34) ($p$ = 0.016) for menarche <12 years and 1.19 (1.07, 1.34) ($p$ = 0.024) for menarche >14 years compared to 13 years; 0.85 (0.74, 0.98) ($p$ = 0.026) for ever been pregnant; 1.43 (1.26, 1.62) ($p$ < 0.001) for age at first live birth <21 compared to 25 to 26 years; 0.82 (0.71, 0.94) ($p$ = 0.006) for each abortion; 1.32 (1.15, 1.51) ($p$ = 0.008) for natural menopause at <47 compared to 50 years; 1.12 (1.01, 1.25) ($p$ = 0.039) for hysterectomy; 2.35 (1.06, 5.23) ($p$ = 0.037) for hysterectomy with previous oophorectomy; and 0.80 (0.72, 0.88) ($p$ < 0.001) for oral contraceptive pills use. The U-shaped associations between the number of children and the risk of dementia were similar for both sexes: Compared with those with 2 children, for those without children, the multiple adjusted HR ((95% CIs), $p$-value) was 1.18 (1.04, 1.33) ($p$ = 0.027) for women and 1.10 (0.98, 1.23) ($p$ = 0.164) for men, and the women-to-men ratio of HRs was 1.09 (0.92, 1.28) ($p$ = 0.403); for those with 4 or more children, the HR was 1.14 (0.98, 1.33) ($p$ = 0.132) for women and 1.26 (1.10,

**Data Availability Statement:** The UK Biobank resources are available upon reasonable request and can be accessed through applications on their website (https://www.ukbiobank.ac.uk/enable-your-research/apply-for-

access), and by contacting access@ukbiobank.ac.uk.

**Funding:** JG is supported by the Scientia PhD scholarship from the University of New South Wales, Sydney, Australia (https://www.scientia.unsw.edu.au/). SAEP is supported by a UK Medical Research Council Skills Development Fellowship [grant number: MR/P014550/1] (https://mrc.ukri.org/). MW is supported by an Australian National Health and Medical Research Council Investigator Grant [grant number APP1174120] and Program Grant [grant number: APP1149987] (https://www.nhmrc.gov.au/). The funders had no role in study design, data collection and analysis, decision to publish, or preparation of the manuscript.

**Competing interests:** I have read the journal's policy and the authors of this manuscript have the following competing interests: MW does consultancy for Amgen, Freeline and Kirin outside the submitted work.

**Abbreviations:** BMI, body mass index; CI, confidence interval; COSMIC, Cohort Studies of Memory in an International Consortium; CPRD, Clinical Practice Research Datalink; HR, hazard ratio; HRT, hormone replacement therapy; ICD-10, International Classification of Diseases-10th Revision; KP, Kaiser Permanente; MICE, Multivariate Imputation by Chained Equations; RHR, ratio of hazard ratio; SD, standard deviation; SES, socioeconomic status; STROBE, Strengthening the Reporting of Observational Studies in Epidemiology; WHIS, Women's Health Initiative Study.

1.45) ($p = 0.003$) for men, and the women-to-men ratio of HRs was 0.93 (0.76, 1.14) ($p = 0.530$). There was evidence that hysterectomy (HR, 1.31 (1.09, 1.59), $p = 0.013$) and oophorectomy (HR, 1.39 (1.08, 1.78), $p = 0.002$) were associated with a higher risk of dementia among women of relatively lower SES only. Limitations of the study include potential residual confounding and self-reported measures of reproductive factors, as well as the limited representativeness of the UK Biobank population.

## Conclusions

In this study, we observed that some reproductive events related to shorter cumulative endogenous estrogen exposure in women were associated with higher dementia risk, and there was a similar association between the number of children and dementia risk between women and men.

## Author summary

### Why was this study done?

- Dementia rates are increasing around the world, with some studies reporting a higher incidence in women than men.
- Evidence on the associations between reproductive factors and the risk of dementia remains uncertain.
- Hormone use in women and their associations with dementia risk remain unclear.

### What did the researchers do and find?

- Early and late menarche, younger age at first birth, and hysterectomy were associated with a greater dementia risk; ever been pregnant, ever had an abortion, longer reproductive span, and later menopause were associated with a lower risk of all-cause dementia, after controlling for key confounders, using data from the UK Biobank.
- Hysterectomy, specifically hysterectomy without concomitant oophorectomy or with a previous oophorectomy, was associated with greater dementia risk.
- Use of oral contraceptive pills was associated with a lower dementia risk.
- The U-shaped associations between number of children and dementia appeared similar for both sexes.

### What do these findings mean?

- In this study, we observed that certain reproductive factors are associated with greater risk of dementia. Future work is needed to understand whether this is related to the fact that these factors may be associated with shorter cumulative exposure to endogenous estradiol.

- Findings suggest that risk variation in women may not be associated with factors associated with childbearing because a similar U-shaped pattern was observed between number of children fathered and dementia risk among men.

## Introduction

The dementia epidemic confronts the world as a major challenge, with extensive impact on individuals, carers, families, and societies at large [1,2]. Fifty million people live with dementia globally, and this number is projected to triple by 2050 [1]. There is no effective course-modifying treatment for dementia to date [2]; mitigation and modification of risk factors, therefore, present opportunities to reduce the burden associated with dementia at a population level [2].

The age-standardised global prevalence and death rates for dementia were estimated to be higher in women than men [3]. While the risk of developing dementia increases with age, the extent to which the female predominance is simply due to women's longer life span remains far from conclusive, and female-specific reproductive factors may be able to explain these sex disparities [4,5].

Several endogenous estrogen changes occur throughout a woman's reproductive life. Estradiol (E2) is the most predominant form of estrogen during reproductive life (from menarche to menopause) [6], and estriol (E3) is the primary estrogen during pregnancy [7]. Exogenous hormone use, such as oral contraceptives during reproductive years, and hormone replacement therapy (HRT) in later life can also influence estrogen level. Few studies evaluated the long-term effect of reproductive factors on dementia risk, such that conclusions remain uncertain, and the putative mechanisms are not well understood [4,8–13].

This study examined the reproductive factors and exogenous hormone use in relation to the risk of incident all-cause dementia in women in the UK Biobank. We assessed whether the associations between these factors and the risk of dementia vary by age, socioeconomic status (SES), smoking status, and body mass index (BMI). In addition, we included men to compare the association between number of children fathered and the risk of all-cause dementia, with the association in female counterparts.

## Methods

### Study design

No prospective analysis plan was specifically designed for the current study, although the framework for the design was drawn from the analyses carried out in a previous study, which examined the associations between reproductive factors and cardiovascular diseases in the UK Biobank [14], with prespecified subgroup analyses to assess effect modification.

The study design was further augmented by including broader selection and exposure categories, such as exogenous hormone use, and the timing of HRT in relation to menopause, as these are critical aspects to consider which may implicate the risk of dementia based on previous literature [4,8–13].

Post hoc sensitivity analyses included death as a competing risk in estimating the associations between reproductive factors and dementia risk, given that dementia requiring a follow-up measurement which death may preclude. Combinations of reproductive factors, rather than examined in isolation, were also included in the ancillary analyses. Last, the analyses were reweighted according to the social structure of the population in general [15].

## Study population

The UK Biobank is a prospective population-based cohort, recruited over 500,000 (aged 40 to 69 years) women and men between 2006 and 2010 [16]. Individuals were invited to attend one of the 22 centres across the United Kingdom for baseline assessment, which included questionnaires soliciting information on lifestyle, medical history, and reproductive history. Physical measurements were collected, and a blood sample was taken. Written informed consent was obtained for all UK Biobank participants electronically. UK Biobank has obtained Research Tissue Bank approval from its governing Research Ethics Committee, as recommended by the National Research Ethics Service. This research has been conducted using the UK Biobank Resource (application No. 2495). Permission to use the UK Biobank Resource was approved by the access subcommittee of the UK Biobank Board.

## Measurement of reproductive factors

Self-reported reproductive factors included in this study were age at menarche, pregnancy history, number of live births, age at first live birth, number of stillbirths, number of miscarriages, number of abortions, reproductive life span, (age at) natural menopause, (age at) hysterectomy, and (age at) oophorectomy. Exogenous hormone exposures included oral contraceptive pills use, age started using oral contraceptive pills, use of HRT, age at HRT initiation, and duration of HRT use. Early menarche was defined as age at the first menstrual period before 12 years of age. Early natural menopause was defined as the permanent absence of a menstrual period before 47 years of age. The reproductive life span was defined as the difference between the age at natural menopause and the age at menarche. Age at hysterectomy and oophorectomy was used to determine the timing of these procedures. Age at natural or artificial menopause, used to examine the risk associated in relation to the timing of HRT use, was defined as age at natural menopause, age at hysterectomy, or age at oophorectomy, whichever took place first. The number of children fathered was also recorded for men and was thus analysed here.

## Study endpoint

The primary endpoint in this study was incident (i.e., fatal or nonfatal) all-cause dementia, as defined by the UK Biobank Outcome Adjudication Group, using the International Classification of Diseases-10th Revision (ICD-10) codes A81.0, F00, F01, F02, F03, F05, G30, G31.0, G31.1, G31.8, and I67.3 [17]. Hospital inpatient data from England, Scotland, and Wales, as well as the national death registers, were used to identify the date of the first known dementia after the date of baseline assessment. Follow-up for all participants started at the entry to the study, with data from the death registers and hospital inpatient data ended on November 30, 2020 or when fatal, nonfatal all-cause dementia, or death was recorded.

## Covariates

Social deprivation was determined from the Townsend deprivation index [18]. Townsend deprivation index measures area deprivation, comprised of 4 domains including information about unemployment, car ownership, household overcrowding, and owner occupation, derived from the national census data, with higher scores indicate higher levels of social deprivation. The Townsend deprivation index was calculated for participant immediately prior to joining UK Biobank, based on the preceding national census output areas, in which the participant's postcode is located. Smoking status was self-reported and categorised as never, former, or current smokers. Systolic blood pressure was taken at study baseline using the Omron HEM-7015IT digital blood pressure monitor as the mean of 2 sitting measures. BMI was

calculated as the weight of the individual in kilogrammes, measured using the Tanita BC-418 MA body composition analyser, divided by the square of the individual's standing height in metres. Diabetes status was self-reported: If the age at diagnosis was younger than 30, and the participant was using insulin, they were classified as type 1 diabetes, otherwise as type 2 diabetes. Total cholesterol was measured using the Beckman Coulter AU580. Self-reported medication use was also recorded.

## Statistical analysis

The present analyses excluded participants with prevalent dementia at baseline ($N = 263$). Baseline characteristics are presented as mean with standard deviation (SD) for continuous variables and number with percentage for categorical variables.

Sex-specific crude incidence rates of dementia were estimated using Poisson regression models, with a log offset for person-years. We estimated the unadjusted and multiple adjusted rates for dementia per 10,000 person-years in all risk factor categories. Unadjusted and multiple adjusted models included age at study entry, SES, ethnicity, smoking status, systolic blood pressure, BMI, history of diabetes mellitus, total cholesterol, antihypertensive drugs, and lipid-lowering drugs.

The associations between each reproductive factors and dementia were assessed using Cox proportional hazard regression models that estimated the hazard ratios (HRs) with accompanying 95% confidence intervals (CIs) and $p$-values. When more than 2 groups were compared, the 95% CIs were estimated using floating absolute risks [19]. Covariate adjustments were the same as those made in the Poisson models. The association between the number of children fathered and dementia was assessed in men, fitted with the same set of covariates, to make a direct comparison with the number of live births in women, and the interaction term between the number of children and sex was used to obtain the women-to-men ratio of hazard ratios (RHRs) [20].

A series of models to explore combinations of reproductive factors were also constructed. These were (A) a combination of pregnancy related factors: number of live births, stillbirths, miscarriages, and abortions; and (B) a combination of factors throughout the life span including age at menarche, parous versus not, hysterectomy and or oophorectomy, HRT use, and contraceptive pill use.

Multiple adjusted restricted cubic splines (with kernel density plots) were constructed to assess the shape of continuous reproductive factors associated with dementia risk. The top and bottom 2.5% of the distributions, where precision is poor, were excluded, with the median value of the distribution taken as the reference.

Predefined subgroup analyses were conducted by age group (categorised as ≥65 versus <65 years, to yield an approximately equal number of events in each group), social deprivation (determined using the Townsend deprivation index at or below versus above the national median (−0.56)), smoking status (ever versus never smoker), and BMI (>25 kg/m$^2$ versus ≤25 kg/m$^2$), to examine the effect modifications by these characteristics. The interaction term was fitted between the exposure of interest and the prespecified subgroup to obtain the $p$-value. We examined social deprivation in finer detail by grouping the Townsend deprivation score into fifths based on nationally derived cutoffs, given the heterogeneity observed in subgroup analysis by the predefined 2-level SES. The lowest fifth contained the 20% least socially disadvantaged, and the highest fifth contained the 20% most socially disadvantaged, with the interaction term fitted between the exposure of interest and Townsend fifths, and $p$ for heterogeneity were presented. In addition, the results were weighted according to the social structure of the population in general: Weighted HR were calculated according to equal weights of each

stratified HR by fifths of Townsend score, since the national Townsend fifths, by definition, contain an equal number of people.

We excluded women who underwent hysterectomy or oophorectomy for sensitivity analysis and examined the risk of dementia associated with age at menarche, reproductive years, and age at natural menopause.

We also constructed Fine and Gray competing risk models [21] and multinomial regression models [22], to compare with results from Cox proportional hazards regression models, as sensitivity analysis. These methods will enable death to be accounted for as a competing risk for dementia, given that death may preclude dementia from occurring. The competing risk models, which estimated subdistribution HRs, were conducted for all-cause dementia accounting for all-cause mortality as a competing risk, incorporating time to event data. For multinomial regression models, the odds ratios were produced, with outcomes specified as (0) had neither dementia nor died by the last follow-up; (1) all-cause dementia by the last follow-up; and (2) all-cause mortality preceding all-cause dementia by the last follow-up. Time to event is not specified in multinomial models.

All main analyses were performed on complete case data using R version 4.1.0 (RStudio Team (2021). RStudio: Integrated Development for R. RStudio, PBC, Boston) and Stata 17.0 (StataCorp. 2021. Stata Statistical Software: Release 17. College Station, TX: StataCorp LLC).

Further, missing data were imputed using Multivariate Imputation by Chained Equations (MICE), with 5 iterations. The multiple adjusted results from Cox regression models with imputed data were subsequently compared with the complete case analyses.

This study is reported as per the Strengthening the Reporting of Observational Studies in Epidemiology (STROBE) guideline (S1 Checklist).

## Results

Over a median of 11.8 years follow-up, 1,866 cases of incident dementia were recorded among 273,240 women. At the study baseline, the mean age of women was 56 years; the mean age at natural menarche and age at first live birth was 13 and 26 years, respectively; 85% reported they have been pregnant at least once, and 44% reported having 2 children. For natural menopause, 61% of the women were postmenopausal, and the mean age at natural menopause was 50 years. For surgical-induced menopause, the percentage of women who reported a history of hysterectomy and oophorectomy were 19% and 8%, respectively; 81% reported ever used oral contraceptive pills, and 38% reported ever used HRT, with a mean age of 47 years for HRT initiation and a mean duration of 6.3 years.

Among 228,957 men, 2,202 incident cases of dementia were recorded. The mean age for men at baseline was 57 years, with 41% reported fathering 2 children (Table 1).

Missing data for each reproductive variable of interest were reported in the Supporting information (S1 Table).

### Dementia rates

The crude incidence rate for dementia was 5.88 (95% CI (5.62, 6.16)) for women and 8.42 (8.07, 8.78) for men per 10,000 person-years.

The multiple adjusted rates of dementia per 10,000 person-years (95% CI) were the highest among those with shorter reproductive span (<33 years: 8.15 (6.87, 9.42)) and earlier age at natural menopause (<47 years: 8.85 (7.66, 10.04)) (Table 2).

Among men, the multiple adjusted rates of all-cause dementia per 10,000 person-years (95% CI) were 8.65 (7.68, 9.62), 7.69 (6.63, 8.75), 7.95 (7.39, 8.51), 8.56 (7.70, 9.43), and 10.04 (8.67, 11.42) and for those who fathered none, 1, 2, 3, and 4 or more children, respectively.

**Table 1. Baseline characteristics of study participants in the UK Biobank.**

| | Women (*n* = 273,240) | Men (*n* = 228,957) |
|---|---|---|
| **Dementia, *n* %** | **1,866 (0.7)** | **2,202 (1.0)** |
| Age, years | 56.3 (8.0) | 56.7 (8.0) |
| Social deprivation, % | | |
| Higher (≤−0.56 on Townsend deprivation index) | 183,388 (67.1) | 151,860 (66.3) |
| Lower (>−0.56 on Townsend deprivation index) | 66,740 (24.4) | 58,259 (25.4) |
| Townsend fifths, % | | |
| First (<−2.938 on Townsend deprivation index (least disadvantaged)) | 100,950 (36.9) | 84,842 (37.1) |
| Second (≥−2.938, <1.531 on Townsend deprivation index) | 56,567 (20.7) | 46,147 (20.2) |
| Third (≥−1.531, <0.170 on Townsend deprivation index) | 41,229 (15.1) | 33,314 (14.6) |
| Fourth (≥0.170, 2.448 on Townsend deprivation index) | 36,888 (13.5) | 30,431 (13.3) |
| Fifth (≥2.448 on Townsend deprivation index (most disadvantaged)) | 37,279 (13.6) | 33,927 (14.8) |
| Ethnicity, % | | |
| White | 257,304 (94.2) | 215,104 (93.9) |
| Other | 14,672 (5.8) | 12,344 (5.4) |
| Smoking status, % | | |
| Never smoker | 161,965 (59.3) | 111,401 (48.7) |
| Former smoker | 85,407 (31.3) | 87,534 (38.2) |
| Current smoker | 24,356 (8.9) | 28,590 (12.5) |
| Blood pressure | | |
| Systolic blood pressure (mean (SD)) | 135.3 (19.2) | 140.9 (17.5) |
| Diastolic blood pressure (mean (SD)) | 80.7 (10.0) | 84.1 (10.0) |
| BMI (mean (SD)) | 27.1 (5.2) | 27.8 (4.2) |
| Diabetes, % | | |
| Type 1 diabetes[a] | 564 (0.2) | 652 (0.3) |
| Type 2 diabetes | 9,945 (3.6) | 15,514 (6.8) |
| Total cholesterol (mean (SD)) | 5.87 (1.1) | 5.48 (1.1) |
| Antihypertensive drugs | 38,405 (14.1) | 47,965 (20.9) |
| Lipid-lowering drugs | 29,502 (10.8) | 45,730 (20.0) |
| Age at menarche, years | 13.0 (1.6) | - |
| Ever pregnant, % | 231,352 (84.7) | - |
| Number of children, % | | |
| None | 51,079 (18.7) | 47,098 (20.6) |
| 1 | 36,457 (13.3) | 28,635 (12.5) |
| 2 | 119,113 (43.6) | 94,263 (41.2) |
| 3 | 48,270 (17.7) | 38,128 (16.7) |
| 4 or more | 17,493 (6.4) | 16,562 (7.2) |
| Age at first live birth, years | 25.9 (5.1) | - |
| Number of miscarriages, % | | |
| None | 171,511 (62.8) | - |
| 1 | 40,042 (14.7) | - |
| 2 or more | 15,876 (5.8) | - |
| Number of stillbirths, % | | |
| None | 220,585 (80.7) | - |
| 1 | 6,072 (2.2) | - |
| 2 or more | 963 (0.4) | - |
| Number of abortions, % | | |
| None | 189,105 (69.2) | - |

*(Continued)*

**Table 1.** (Continued)

| | Women (*n* = 273,240) | Men (*n* = 228,957) |
|---|---|---|
| 1 | 30,536 (11.2) | - |
| 2 or more | 7,349 (2.7) | - |
| Number of reproductive years | 37.3 (4.8) | - |
| Menopause | | |
| Natural menopause | | |
| Postmenopausal, % | 165,856 (60.7) | - |
| Age at menopause, years | 50.3 (4.5) | - |
| Surgical menopause | | |
| History of hysterectomy, % | 51,226 (18.7) | - |
| Age at hysterectomy, years | 43.9 (8.0) | - |
| History of oophorectomy, % | 21,935 (8.0) | - |
| Age at oophorectomy, years | 47.4 (7.8) | - |
| History of both hysterectomy and oophorectomy, % | 20,901 (7.6) | |
| Exogenous hormone use | | |
| Oral contraceptive pills use | | |
| Ever used oral contraceptive pills, % | 220,344 (80.6) | - |
| Age first taken oral contraceptive pills, years | 21.5 (4.7) | - |
| HRT use | | |
| Ever used HRT, % | 104,133 (38.1) | - |
| Age initiated HRT, years | 47.4 (5.4) | - |
| HRT duration, years | 6.3 (5.3) | - |

[a]Defined as diagnosis before the age of 30 and receiving insulin treatment.

SD, standard deviation; HRT, hormone replacement therapy.

These rates were higher among men when compared with number of live births reported in women across all the categories (Table 2).

## Age at menarche

Overall, the relationship between age at menarche and dementia appeared to be U shaped (Table 3, Fig 1A): The multiple adjusted HRs (95% CI) of the age at menarche <12 associated with dementia was 1.20 (1.08, 1.34) (*p* = 0.016) and at the age of >14 was 1.19 (1.07, 1.34) (*p* = 0.024), compared to women who had their menarche at 13.

## Parity-related factors

The HR for dementia who had ever been pregnant was 0.85 (0.74, 0.98) (*p* = 0.026) compared with never pregnant. Younger age at first live birth was associated with a higher dementia risk, with the HR for <21 years at first live birth was 1.43 (1.26, 1.62) (*p* < 0.001)) compared with first live birth at 25 to 26 years (Table 3, Fig 1B). Compared with women who never had an abortion, the HR for dementia in women who had 2 or more abortions was 0.34 (0.18, 0.64) (*p* < 0.001). Stillbirth and miscarriage were not associated with dementia risk.

## Number of children

Compared with those who had 2 children, the associations between the number of children and dementia were similar for women and men and appeared to be U shaped (Fig 2, S2 Table):

**Table 2. Unadjusted and multiple adjusted rates of incident dementia for reproductive risk factors in women.**

| Reproductive factor | Unadjusted rates/10,000 person-years (95% CI) | Multiple adjusted rates/10,000 person-years (95% CI)[a] |
|---|---|---|
| Age at menarche | | |
| <12 | 6.27 (5.64, 6.90) | 6.31 (5.65, 7.00) |
| 12 | 5.69 (5.07, 6.30) | 5.64 (4.98, 6.29) |
| 13 | 4.96 (4.46, 5.47) | 5.26 (4.70, 5.83) |
| 14 | 5.51 (4.92, 6.10) | 5.13 (4.54, 5.72) |
| >14 | 6.58 (5.89, 7.28) | 6.31 (5.60, 7.01) |
| Ever been pregnant | | |
| No | 5.12 (4.48, 5.77) | 6.69 (5.80, 7.58) |
| Yes | 6.00 (5.71, 6.30) | 5.72 (5.42, 6.01) |
| Number of live births | | |
| 0 | 4.80 (4.24, 5.36) | 6.46 (5.67, 7.26) |
| 1 | 5.16 (4.47, 5.84) | 6.01 (5.18, 6.84) |
| 2 | 5.58 (5.19, 5.97) | 5.52 (5.11, 5.93) |
| 3 | 6.99 (6.30, 7.68) | 5.72 (5.11, 6.33) |
| 4 or more | 9.41 (8.07, 10.75) | 6.35 (5.37, 7.33) |
| Parous | | |
| No | 4.80 (4.24, 5.36) | 6.46 (5.67, 7.26) |
| Yes | 6.12 (5.81, 6.42) | 5.73 (5.43, 6.03) |
| Age at first live birth | | |
| <21 | 8.31 (7.37, 9.24) | 7.15 (6.28, 8.03) |
| 21 to 22 | 8.32 (7.32, 9.31) | 6.17 (5.39, 6.95) |
| 23 to 24 | 7.87 (6.98, 8.76) | 6.32 (5.56, 7.09) |
| 25 to 26 | 5.33 (4.62, 6.04) | 5.01 (4.29, 5.72) |
| 27 to 29 | 4.87 (4.26, 5.47) | 5.79 (5.04, 6.55) |
| >29 | 3.75 (3.25, 4.25) | 5.54 (4.74, 6.34) |
| Number of miscarriages | | |
| 0 | 6.12 (5.78, 6.47) | 5.99 (5.64, 6.35) |
| 1 | 5.35 (4.68, 6.01) | 5.38 (4.66, 6.10) |
| 2 or more | 5.77 (4.67, 6.87) | 6.12 (4.88, 7.36) |
| Number of stillbirths | | |
| 0 | 5.86 (5.57, 6.16) | 5.88 (5.56, 6.19) |
| 1 | 9.31 (7.05, 11.58) | 6.72 (4.95, 8.48) |
| 2 or more | 11.01 (4.78, 17.24) | 7.41 (2.56, 12.25) |
| Number of abortions | | |
| 0 | 6.41 (6.08, 6.74) | 6.02 (5.69, 6.36) |
| 1 | 4.35 (3.67, 5.04) | 5.79 (4.82, 6.77) |
| 2 or more | 1.54 (0.70, 2.37) | 2.04 (0.77, 3.31) |
| Reproductive years | | |
| <33 | 8.60 (7.36, 9.83) | 8.15 (6.87, 9.42) |
| 33 to 35 | 7.56 (6.45, 8.67) | 8.01 (6.77, 9.26) |
| 36 to 37 | 6.10 (5.17, 7.03) | 6.38 (5.36, 7.41) |
| 38 to 39 | 5.20 (4.40, 6.01) | 5.47 (4.58, 6.35) |
| 40 to 42 | 5.82 (4.99, 6.64) | 5.52 (4.68, 6.36) |
| >42 | 7.72 (6.43, 9.01) | 6.43 (5.31, 7.56) |
| Age at natural menopause | | |
| <47 | 8.98 (7.85, 10.11) | 8.85 (7.66, 10.04) |

*(Continued)*

**Table 2.** (Continued)

| Reproductive factor | Unadjusted rates/10,000 person-years (95% CI) | Multiple adjusted rates/10,000 person-years (95% CI)[a] |
|---|---|---|
| 47 to 49 | 6.38 (5.39, 7.36) | 7.20 (6.01, 8.38) |
| 50 | 6.99 (5.97, 8.02) | 6.73 (5.69, 7.76) |
| 51 to 52 | 5.34 (4.54, 6.13) | 5.38 (4.51, 6.25) |
| 53 to 54 | 4.90 (3.97, 5.83) | 5.12 (4.09, 6.14) |
| >54 | 7.66 (6.58, 8.74) | 6.21 (5.29, 7.13) |
| Hysterectomy | | |
| No | 5.64 (5.31, 5.96) | 5.64 (5.31, 5.96) |
| Yes | 6.32 (5.75, 6.89) | 6.32 (5.75, 6.89) |
| Oophorectomy | | |
| No | 5.52 (5.24, 5.79) | 5.69 (5.39, 5.98) |
| Yes | 8.61 (7.47, 9.76) | 6.04 (5.18, 6.90) |
| Ever taken oral contraceptive pills | | |
| No | 11.23 (10.38, 12.09) | 6.79 (6.23, 7.36) |
| Yes | 4.60 (4.34, 4.87) | 5.37 (5.04, 5.70) |
| Ever used HRT | | |
| No | 4.33 (4.04, 4.62) | 5.83 (5.40, 6.25) |
| Yes | 8.28 (7.77, 8.79) | 5.77 (5.39, 6.15) |

[a]Analyses were adjusted for age, Townsend index, ethnicity, smoking status, systolic blood pressure, BMI, diabetes, total cholesterol, antihypertensive drugs, and lipid-lowering drugs.

CI, confidence interval; HRT, hormone replacement therapy.

for instance, for those who had no children, the adjusted HR was 1.18 (1.04, 1.33) ($p$ = 0.027) for women and 1.10 (0.98, 1.23) ($p$ = 0.164) for men, with the women-to-men ratio of HRs being 1.09 (0.92, 1.28) ($p$ = 0.403); for those who had 4 or more children, the adjusted HR was 1.14 (0.98, 1.33) ($p$ = 0.132) for women and 1.26 (1.10, 1.45) ($p$ = 0.003) for men, with the women-to-men ratio of HRs being 0.93 (0.76, 1.14) ($p$ = 0.530). Unadjusted results by predefined subgroups were also presented in the Supporting information (S3 Table).

## Menopause-related factors

A longer reproductive life span and an older age at natural menopause had inverse log-linear associations with dementia risk (Table 3, Fig 1C and 1D). The HR of dementia associated with menopause before the age of 47 was 1.32 (1.15, 1.51) ($p$ = 0.008) compared to women who had their menopause at the age of 50. For women who reported a history of hysterectomy, the HR for dementia was 1.12 (1.01, 1.25) ($p$ = 0.039) compared with women who never had a hysterectomy. Oophorectomy was not significantly associated with dementia; the HR was 1.07 (0.92, 1.24) ($p$ = 0.413). Younger age at hysterectomy was associated with increased dementia risk (Fig 1E), whereas the relationship for age at oophorectomy and dementia risk appear to be U shaped (Fig 1F).

Regarding the timing of hysterectomy and oophorectomy, in comparison to women who never underwent hysterectomy or oophorectomy, women who had hysterectomy after oophorectomy had an increased risk (HR, 2.35 (1.06, 5.23), $p$ = 0.037), while the other variations for the timing of the procedures did not appear to be associated with dementia risk (S4 Table).

**Table 3. Unadjusted and multiple adjusted HRs for the risk of dementia associated with reproductive factors in women.**

| Reproductive factor | No. of events | Unadjusted HR (95% CI) | p-Value | Multiple adjusted HR (95% CI) [a] | p-Value |
|---|---|---|---|---|---|
| Age at menarche | | | | | |
| <12 | 385 | 1.26 (1.14, 1.40) | <0.001 | 1.20 (1.08, 1.34) | 0.016 |
| 12 | 331 | 1.15 (1.03, 1.28) | 0.072 | 1.07 (0.95, 1.20) | 0.389 |
| 13 (ref) | 372 | 1.00 (0.90, 1.11) | - | 1.00 (0.90, 1.11) | - |
| 14 | 334 | 1.11 (1.00, 1.24) | 0.167 | 0.97 (0.87, 1.09) | 0.718 |
| >14 | 341 | 1.33 (1.19, 1.48) | <0.001 | 1.19 (1.07, 1.34) | 0.024 |
| Ever been pregnant | 1,613 | 1.17 (1.02, 1.34) | 0.022 | 0.85 (0.74, 0.98) | 0.026 |
| Number of live births | | | | | |
| 0 | 284 | 0.86 (0.75, 0.98) | 0.338 | 1.18 (1.04, 1.33) | 0.027 |
| 1 | 218 | 0.93 (0.79, 1.06) | 0.312 | 1.09 (0.95, 1.25) | 0.276 |
| 2 (ref) | 773 | 1.00 (0.93, 1.07) | - | 1.00 (0.93, 1.08) | - |
| 3 | 392 | 1.25 (1.15, 1.35) | <0.001 | 1.03 (0.93, 1.15) | 0.641 |
| 4 or more | 190 | 1.69 (1.55, 1.83) | <0.001 | 1.14 (0.98, 1.33) | 0.132 |
| Parous versus not | 1,573 | 1.27 (1.12, 1.44) | <0.001 | 0.88 (0.77, 1.01) | 0.061 |
| Age at first live birth | | | | | |
| <21 | 305 | 1.56 (1.40, 1.75) | <0.001 | 1.43 (1.26, 1.62) | <0.001 |
| 21 to 22 | 270 | 1.56 (1.39, 1.76) | <0.001 | 1.23 (1.08, 1.40) | 0.034 |
| 23 to 24 | 302 | 1.48 (1.32, 1.65) | <0.001 | 1.26 (1.12, 1.42) | 0.015 |
| 25 to 26 (ref) | 217 | 1.00 (0.88, 1.14) | - | 1.00 (0.87, 1.15) | - |
| 27 to 29 | 251 | 0.91 (0.81, 1.03) | 0.324 | 1.16 (1.02, 1.32) | 0.129 |
| >29 | 213 | 0.70 (0.62, 0.81) | <0.001 | 1.11 (0.96, 1.29) | 0.296 |
| Per additional year of age at first live birth | - | 0.94 (0.93, 0.95) | <0.001 | 0.98 (0.97, 1.00) | 0.006 |
| Number of miscarriages | | | | | |
| 0 (ref) | 1,220 | 1.00 (0.95, 1.06) | - | 1.00 (0.94, 1.06) | - |
| 1 | 249 | 0.87 (0.77, 0.99) | 0.050 | 0.90 (0.78, 1.03) | 0.147 |
| 2 or more | 106 | 0.94 (0.78, 1.14) | 0.571 | 1.02 (0.83, 1.25) | 0.847 |
| Miscarriage versus not | 355 | 0.89 (0.79, 1.00) | 0.060 | 0.93 (0.82, 1.06) | 0.274 |
| Per miscarriage | - | 1.03 (0.97, 1.10) | 0.308 | 1.01 (0.94, 1.08) | 0.816 |
| Number of stillbirths | | | | | |
| 0 (ref) | 1,503 | 1.00 (0.95, 1.05) | - | 1.00 (0.94, 1.06) | - |
| 1 | 65 | 1.59 (1.25, 2.03) | <0.001 | 1.15 (0.88, 1.49) | 0.312 |
| 2 or more | 12 | 1.90 (1.08, 3.35) | 0.027 | 1.27 (0.66, 2.45) | 0.472 |
| Stillbirth versus not | 77 | 1.63 (1.30, 2.06) | <0.001 | 1.16 (0.91, 1.49) | 0.234 |
| Per stillbirth | - | 1.28 (1.12, 1.45) | <0.001 | 1.10 (0.92, 1.31) | 0.319 |
| Number of abortions | | | | | |
| 0 (ref) | 1,409 | 1.00 (0.95, 1.05) | - | 1.00 (0.94, 1.07) | - |
| 1 | 154 | 0.68 (0.58, 0.80) | <0.001 | 0.97 (0.82, 1.14) | 0.735 |
| 2 or more | 13 | 0.24 (0.14, 0.42) | <0.001 | 0.34 (0.18, 0.64) | <0.001 |
| Abortion versus not | 167 | 0.60 (0.51, 0.70) | <0.001 | 0.87 (0.73, 1.03) | 0.101 |
| Per abortion | - | 0.62 (0.54, 0.71) | <0.001 | 0.82 (0.71, 0.94) | 0.006 |
| Reproductive years | | | | | |
| <33 (ref) | 186 | 1.00 (0.87, 1.15) | - | 1.00 (0.86, 1.17) | - |
| 33 to 35 | 178 | 0.88 (0.76, 1.02) | 0.214 | 0.98 (0.84, 1.15) | 0.878 |
| 36 to 37 | 166 | 0.71 (0.61, 0.83) | 0.001 | 0.78 (0.67, 0.92) | 0.031 |
| 38 to 39 | 160 | 0.60 (0.52, 0.71) | <0.001 | 0.67 (0.57, 0.79) | <0.001 |
| 40 to 42 | 192 | 0.68 (0.59, 0.78) | <0.001 | 0.68 (0.58, 0.79) | <0.001 |
| >42 | 137 | 0.90 (0.76, 1.07) | 0.362 | 0.80 (0.67, 0.95) | 0.056 |

*(Continued)*

**Table 3.** (Continued)

| Reproductive factor | No. of events | Unadjusted HR (95% CI) | p-Value | Multiple adjusted HR (95% CI) [a] | p-Value |
|---|---|---|---|---|---|
| Age at natural menopause | | | | | |
| <47 | 242 | 1.28 (1.13, 1.46) | 0.011 | 1.32 (1.15, 1.51) | 0.008 |
| 47 to 49 | 161 | 0.91 (0.78, 1.06) | 0.382 | 1.07 (0.91, 1.26) | 0.573 |
| 50 (ref) | 180 | 1.00 (0.86, 1.16) | - | 1.00 (0.86, 1.17) | - |
| 51 to 52 | 173 | 0.76 (0.66, 0.88) | 0.010 | 0.80 (0.68, 0.94) | 0.048 |
| 53 to 54 | 107 | 0.70 (0.58, 0.84) | 0.003 | 0.76 (0.62, 0.93) | 0.035 |
| >54 | 193 | 1.10 (0.96, 1.27) | 0.358 | 0.93 (0.80, 1.08) | 0.496 |
| Hysterectomy versus not | 537 | 1.78 (1.61, 1.96) | <0.001 | 1.12 (1.01, 1.25) | 0.039 |
| Oophorectomy versus not | 217 | 1.57 (1.36, 1.81) | <0.001 | 1.07 (0.92, 1.24) | 0.413 |
| Ever taken oral contraceptive pills | 1,180 | 0.41 (0.37, 0.45) | <0.001 | 0.80 (0.72, 0.88) | <0.001 |
| Age started oral contraceptive pills (per year) | - | 1.11 (1.10, 1.12) | <0.001 | 1.01 (1.00, 1.03) | 0.143 |
| Ever used HRT | 997 | 1.91 (1.75, 2.10) | <0.001 | 0.99 (0.90, 1.09) | 0.828 |
| Age started HRT (per year) | - | 0.99 (0.98, 1.01) | 0.306 | 0.96 (0.95, 0.98) | <0.001 |
| Duration of HRT use (per year) | - | 1.03 (1.01, 1.04) | <0.001 | 1.00 (0.98, 1.01) | 0.526 |

[a]Analyses were adjusted for age, Townsend index, ethnicity, smoking status, systolic blood pressure, BMI, diabetes, total cholesterol, antihypertensive drugs, and lipid-lowering drugs.

CI, confidence interval; HR, hazard ratio; HRT, hormone replacement therapy.

For hysterectomy and oophorectomy, there was some evidence of heterogeneity by SES ($p = 0.013$, $p = 0.002$, respectively), such that women of relatively lower SES had increased dementia risk relative to women of higher SES (Table 4).

## Exogenous hormone use

The HR for dementia in women who reported oral contraceptive pill use was 0.80 (0.72, 0.88) ($p < 0.001$) (Table 3). There was no clear evidence of an association between age started using oral contraceptive pills (Fig 1G). There was some evidence of heterogeneity by age, such that the lower risk was only statistically significant in women younger than 65 years at study baseline ($p < 0.001$) (Table 4). There was no evidence of HRT use (0.99 (0.90, 1.09, $p = 0.828$) associated with dementia compared with those never used HRT, although there was evidence of a lower risk in older age at HRT initiation (0.96 (0.95, 0.98) per year, $p < 0.001$) (Table 3, Fig 1H), but no evidence of HRT duration associated with dementia risk (1.00 (0.98, 1.01) per year, $p = 0.526$) (Table 3, Fig 1I).

Further, there was no evidence of dementia risk varied by the timing of HRT initiation in relation to menopause, although those who had menopause with unknown HRT initiation were at an increased risk of dementia compared to those who had menopause without HRT use (HR, 1.49 (1.31, 1.69), $p < 0.001$) (S5 Table).

## Sensitivity analysis

After excluding women who underwent hysterectomy or oophorectomy, the associations for age at menarche, reproductive years, and age at natural menopause in relation to dementia risk were similar to the main results (S6 Table).

Results from the competing risk models (subdistribution HRs) and multinomial regression models (odds ratios), which considered the competing risk of all-cause mortality, showed broadly similar results as the results from the Cox proportional hazards regression models (HRs) (S7 Table).

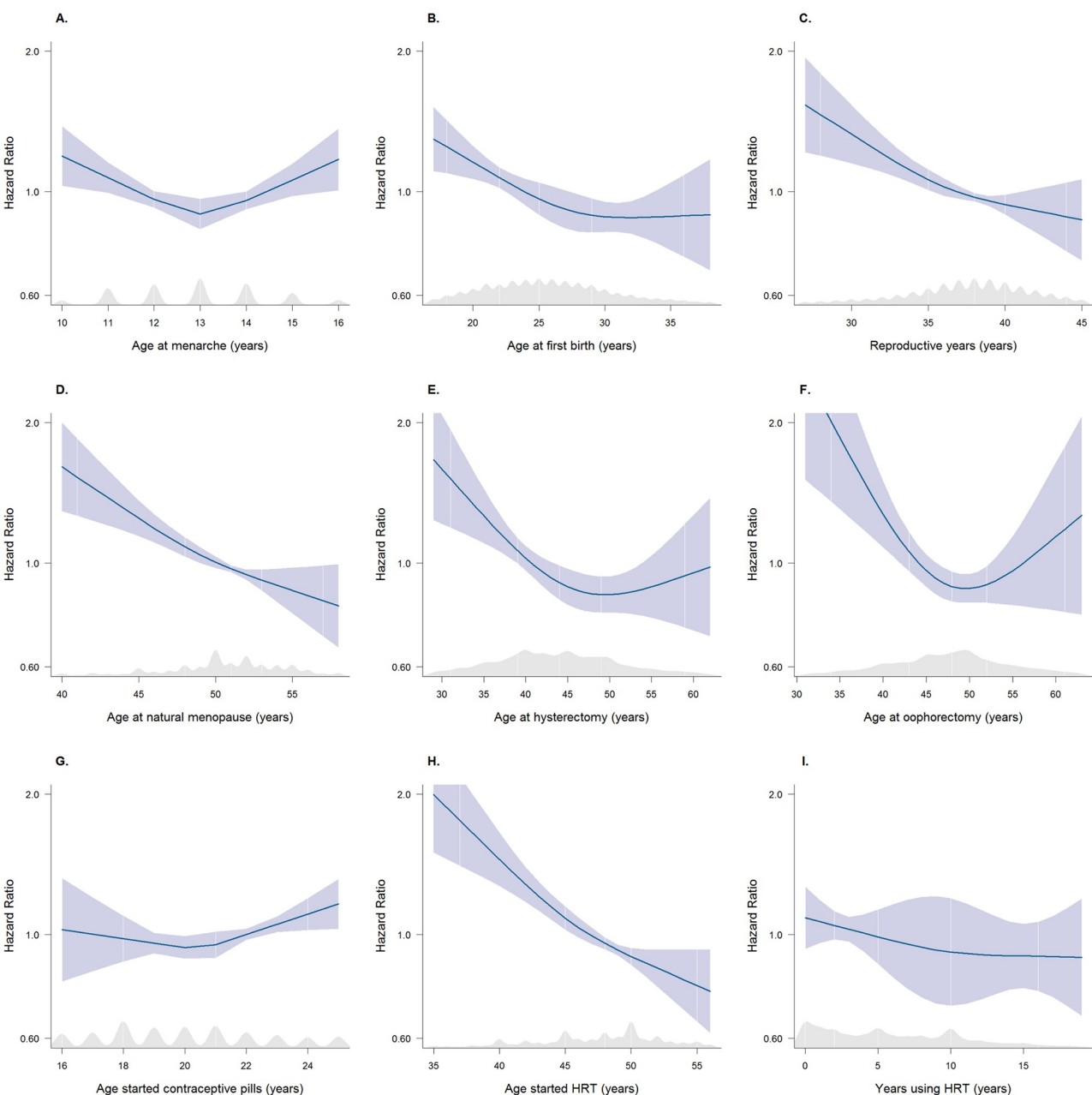

**Fig 1. Multiple adjusted restricted cubic splines (with kernel density plots) showing HRs for the risk of dementia associated with reproductive factors.** The blue line represents the hazard function, and the blue shaded area represents the 95% CIs. The shaded grey region represents the kernel density plot for the distribution of data. After excluding the values from the top and bottom 2.5% of the distribution, with the median value being the reference. Splines adjusted for age, Townsend deprivation index, ethnicity, smoking status, systolic blood pressure, BMI, diabetes, total cholesterol, antihypertensive drugs, and lipid-lowering drugs. **(A)** Restricted cubic spline plot with multiple adjusted HRs (95% CI) for all-cause dementia associated with age at menarche. **(B)** Restricted cubic spline plot with multiple adjusted HRs (95% CI) for all-cause dementia associated with age at first birth. **(C)** Restricted cubic spline plot with multiple adjusted HRs (95% CI) for all-cause dementia associated with reproductive years. **(D)** Restricted cubic spline plot with multiple adjusted HRs (95% CI) for all-cause dementia associated with age at natural menopause. **(E)** Restricted cubic spline plot with multiple adjusted HRs (95% CI) for all-cause dementia associated with age at hysterectomy. **(F)** Restricted cubic spline plot with multiple adjusted HRs (95% CI) for all-cause dementia associated with age at oophorectomy. **(G)** Restricted cubic spline plot with multiple adjusted HRs (95% CI) for all-cause dementia associated with age started contraceptive pills. **(H)** Restricted cubic spline plot with multiple adjusted HRs (95% CI) for all-cause dementia associated with age started HRT. **(I)** Restricted cubic spline plot with multiple adjusted HRs (95% CI) for all-cause dementia associated with years using HRT. BMI, body mass index; CI, confidence interval; HR, hazard ratio; HRT, hormone replacement therapy.

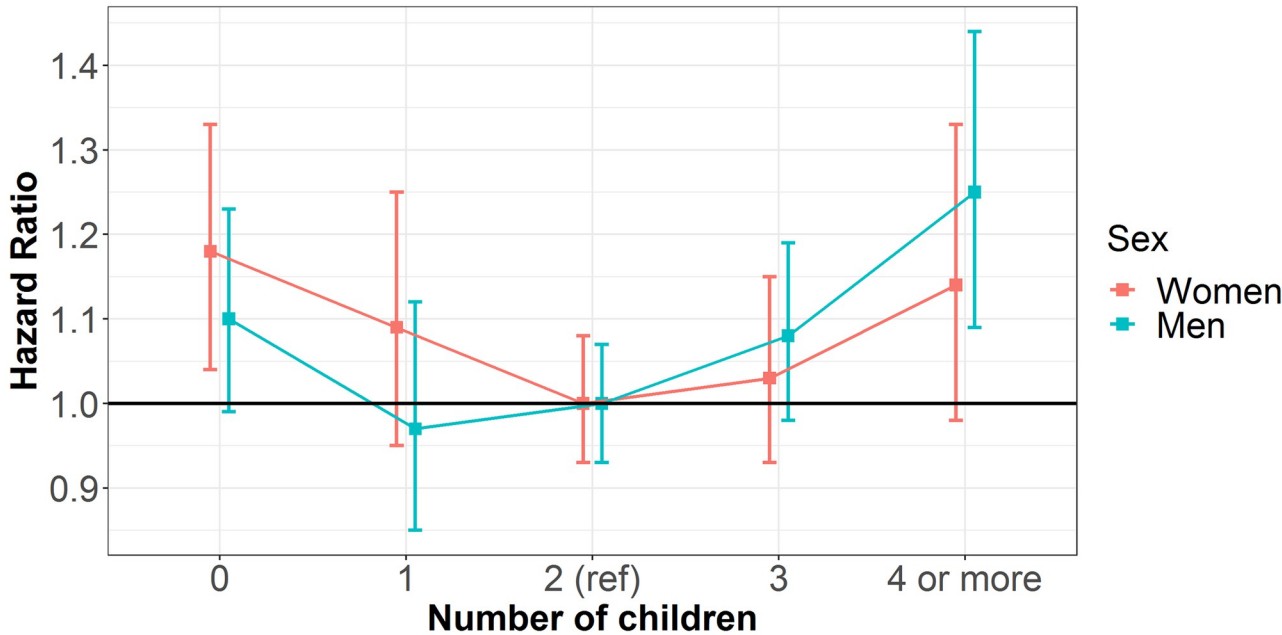

**Fig 2. Multiple adjusted HRs for the risk of dementia associated with number of children for women and men.** The HRs are plotted on a floating absolute scale. The squares represent the HRs, and the bars represent the 95% CIs. Analyses were adjusted for age, Townsend deprivation index, ethnicity, smoking status, systolic blood pressure, BMI, diabetes, total cholesterol, antihypertensive drugs, and lipid-lowering drugs. BMI, body mass index; CI, confidence interval; HR, hazard ratio.

There was some evidence of heterogeneity across Townsend score fifths (S8 Table), including number of children, abortion, oophorectomy, and oral contraceptive pill use in women. After weighting the estimates by Townsend fifths according to the national distribution, the pooled estimates were similar to those in the main analysis but with attenuated the statistical significance of associations as indicated by wider CIs, caused by giving relatively higher weights to the fifths with the least precise estimates (S9 Table). After the results were weighted by Townsend fifths, there was extra imprecision in the weighted HR estimates for some exposures, which occurred where there is considerable difference in estimates (HR and standard errors) across the Townsend fifths; however, this may be a result of small numbers in some exposure categories.

When reproductive factors in the combined model compared with models included these factors individually, there were no major differences, suggesting that there is no substantial effect of confounding (S10 Table).

After imputing missing data, the multiple adjusted results were broadly similar to the results from the complete case analyses (S11 Table).

## Discussion

In this large population-based cohort study, we found several associations between reproductive factors and exogenous use of hormone with dementia risk later in life. Younger age at first live birth, hysterectomy, specifically hysterectomy without concomitant oophorectomy or with a previous oophorectomy, were associated with greater dementia risk. Age at menarche, hysterectomy, and oophorectomy appear to be U shaped for the association with dementia risk. Ever been pregnant, ever had an abortion, longer reproductive span, and older age at natural menopause were associated with lower dementia risk. For exogenous hormone exposures, the

**Table 4. Multiple adjusted HRs for the risk of dementia associated with reproductive factors, by age, SES, smoking, and BMI.**

| Risk factors | Age | | | SES | | |
|---|---|---|---|---|---|---|
| | <65 years | ≥65 years | p-Value | Higher SES | Lower SES | p-Value |
| Early menarche[a] versus not | 1.14 (0.96, 1.36) | 1.15 (0.98, 1.36) | 0.905 | 1.17 (1.00, 1.36) | 1.08 (0.87, 1.35) | 0.820 |
| Age at first live birth per year | 0.98 (0.96, 1.00) | 0.99 (0.97, 1.00) | 0.206 | 0.99 (0.98, 1.01) | 0.97 (0.95, 0.99) | 0.081 |
| Each child[b] | | | | | | |
| Women | 0.97 (0.92, 1.04) | 0.99 (0.94, 1.05) | 0.684 | 0.93 (0.87, 0.98) | 1.06 (1.00, 1.13) | 0.002 |
| Men | 1.05 (1.01, 1.09) | 1.01 (0.96, 1.06) | 0.205 | 1.01 (0.96, 1.06) | 1.05 (1.01, 1.08) | 0.299 |
| Stillbirth versus not | 1.02 (0.66, 1.58) | 1.25 (0.92, 1.69) | 0.531 | 1.16 (0.82, 1.64) | 1.16 (0.78, 1.73) | 0.896 |
| Miscarriage versus not | 0.94 (0.78, 1.14) | 0.92 (0.78, 1.09) | 0.880 | 0.90 (0.76, 1.06) | 1.05 (0.84, 1.32) | 0.219 |
| Abortion versus not | 0.76 (0.59, 0.98) | 0.98 (0.77, 1.24) | 0.179 | 0.84 (0.66, 1.07) | 0.92 (0.70, 1.21) | 0.400 |
| Early menopause[c] versus not | 1.61 (1.28, 2.02) | 1.34 (1.09, 1.64) | 0.146 | 1.27 (1.04, 1.56) | 1.75 (1.34, 2.29) | 0.053 |
| Hysterectomy versus not | 1.16 (0.98, 1.38) | 1.09 (0.95, 1.26) | 0.450 | 0.96 (0.84, 1.11) | 1.31 (1.09, 1.59) | 0.013 |
| Oophorectomy versus not | 1.18 (0.94, 1.48) | 0.99 (0.81, 1.21) | 0.220 | 0.82 (0.66, 1.02) | 1.39 (1.08, 1.78) | 0.002 |
| Oral contraceptive pill use versus not | 0.64 (0.54, 0.76) | 0.92 (0.81, 1.05) | <0.001 | 0.87 (0.76, 1.00) | 0.69 (0.57, 0.83) | 0.176 |
| HRT use versus not | 1.12 (0.96, 1.30) | 0.92 (0.81, 1.05) | 0.069 | 0.95 (0.84, 1.08) | 1.13 (0.94, 1.35) | 0.374 |
| Risk factors | Smoking status | | | BMI | | |
| | Never | Ever | p-Value | ≤25 kg/m² | >25 kg/m² | p-Value |
| Early menarche[a] versus not | 1.07 (0.91, 1.27) | 1.23 (1.03, 1.46) | 0.347 | 1.25 (1.01, 1.55) | 1.10 (0.95, 1.27) | 0.455 |
| Age at first live birth per year | 0.98 (0.96, 0.99) | 0.99 (0.97, 1.01) | 0.318 | 0.98 (0.96, 1.00) | 0.98 (0.97, 1.00) | 0.876 |
| Each child[b] | | | | | | |
| Women | 0.95 (0.90, 1.00) | 1.02 (0.97, 1.09) | 0.098 | 1.00 (0.93, 1.07) | 0.98 (0.94, 1.03) | 0.794 |
| Men | 1.01 (0.95, 1.06) | 1.04 (1.01, 1.08) | 0.340 | 1.06 (1.00, 1.12) | 1.03 (0.99, 1.07) | 0.565 |
| Stillbirth versus not | 1.03 (0.71, 1.48) | 1.31 (0.93, 1.84) | 0.340 | 1.27 (0.82, 1.97) | 1.11 (0.82, 1.50) | 0.553 |
| Miscarriage versus not | 0.82 (0.68, 0.98) | 1.07 (0.89, 1.28) | 0.044 | 0.76 (0.61, 0.96) | 1.02 (0.88, 1.19) | 0.036 |
| Abortion versus not | 0.88 (0.68, 1.14) | 0.86 (0.68, 1.09) | 0.820 | 0.93 (0.70, 1.22) | 0.83 (0.66, 1.03) | 0.521 |
| Early menopause[c] versus not | 1.40 (1.12, 1.74) | 1.49 (1.20, 1.84) | 0.711 | 1.45 (1.13, 1.87) | 1.43 (1.18, 1.74) | 0.735 |
| Hysterectomy versus not | 1.21 (1.05, 1.39) | 1.02 (0.87, 1.20) | 0.113 | 1.03 (0.85, 1.25) | 1.17 (1.03, 1.34) | 0.276 |
| Oophorectomy versus not | 1.14 (0.93, 1.39) | 0.98 (0.78, 1.24) | 0.313 | 1.02 (0.78, 1.34) | 1.08 (0.90, 1.30) | 0.710 |
| Oral contraceptive pill use versus not | 0.79 (0.69, 0.91) | 0.80 (0.68, 0.95) | 0.942 | 0.80 (0.67, 0.95) | 0.80 (0.70, 0.91) | 0.558 |
| HRT use versus not | 0.92 (0.81, 1.05) | 1.09 (0.94, 1.26) | 0.096 | 0.85 (0.72, 1.01) | 1.09 (0.96, 1.23) | 0.043 |

[a]Early menarche was defined as age at first menstrual period before the age of 12 years.

[b]Each live birth in women and each child fathered in men.

[c]Early menopause was defined as the permanent absence of menstrual periods before the age of 47 years.

Analyses were adjusted for age, Townsend index, ethnicity, smoking status, systolic blood pressure, BMI, diabetes, total cholesterol, antihypertensive drugs, and lipid-lowering drugs. p-Values are the interaction between subgroups.

BMI, body mass index; HR, hazard ratio; HRT, hormone replacement therapy; SES, socioeconomic status.

use of oral contraceptive pills was associated with a lower risk of dementia. There was some evidence of heterogeneity by SES for hysterectomy, and oophorectomy, such that the elevated dementia risk associated with these risk factors was confined to women of lower SES. U-shaped associations were found for the number of children and dementia risk, similar for both sexes.

## Surrogates for endogenous hormone exposures

Reproductive events indicating shorter cumulative exposure to estradiol, including later menarche, early natural menopause, shorter reproductive span, and hysterectomy, were all associated with an elevated risk of dementia in our study. However, previous studies on the

relationship between these risk factors have reported mixed results. Consistent with our findings, the Kaiser Permanente (KP) study showed that reproductive events contributing to shorter estradiol exposure were associated with elevated dementia risk [10]. A nationwide study from South Korea also reported these comparable findings [13]. These findings may be driven by the effects of estradiol on brain health; in experimental studies, estradiol has been shown to be correlated with neuronal dendritic spine density [23] as well as reducing apoptosis and inflammation [24]. In contrast, the Gothenburg H70 [9] and Rotterdam study [11] reported that a longer reproductive span and later menopause were associated with greater dementia risk, while the 10/66 study reported no association for reproductive span and dementia [12]. The discrepancy in findings between our study and the Gothenburg H70 and Rotterdam study [9,11] was not due to the exclusion of women who reported hysterectomy or oophorectomy in the Gothenburg H70 and Rotterdam study, as indicated in our sensitivity analyses. However, the studies that reported null or opposite results included older women at study baseline (mean age around 70 years) [11,12] or the women were followed into their late life [9]. Notably, the Gothenburg H70 [9] only found significant associations with dementia for longer reproductive span and older age at menopause, among those with the older onset of dementia (75 years and above) [9]. As such, we hypothesise that the risk exposure in midlife and older life may be different.

Surgically induced menopause (hysterectomy and oophorectomy), when performed before the onset of natural menopause, can cease the secretion of endogenous sex hormones prematurely [25]. The KP study found the dementia risk is greater among women who underwent a hysterectomy [10], which was consistent with our findings. Similarly, a pooled analysis of 2 cohorts found that the risk of cognitive impairment and dementia was higher in women who underwent a hysterectomy, and the risk was even greater in those who had a hysterectomy and bilateral oophorectomy [26]. Moreover, consistent with findings on the timing of surgical menopause procedures in association with cardiovascular disease [14], our study also showed some evidence that there was a greater risk of dementia in those with hysterectomy with a previous oophorectomy. A meta-analysis did not find an overall association between surgical menopause and dementia [27], but surgical menopause before the age of 45 was associated with greater dementia risk [27]. We similarly demonstrated that younger age at hysterectomy and oophorectomy were inversely associated with dementia risk, providing further support that early cessation of hormones may be associated with poorer cognitive outcomes.

When disaggregated by SES, early natural menopause, hysterectomy, and oophorectomy were only associated with a greater risk of dementia in women of relatively low SES. Previous studies reported SES might adversely influence the age at entry to perimenopause [28,29]. Further, social disadvantage can modulate the level of cortisol [30]. During the menopausal transition, increased cortisol level has been associated with vasomotor symptoms and depressed mood [31], which are key determinants for cognitive function [32,33].

## Parity-related factors

Pregnancy induces marked changes in endogenous estrogen levels [7,34], and estrogen can be both neuroprotective or neurotoxic, depending on the concentration [34,35]. A pooled study from the Cohort Studies of Memory in an International Consortium (COSMIC) found that the risk of Alzheimer disease doubled for women who had 4 or more completed pregnancies [34]. Another COSMIC analysis showed that having 5 or more children was associated with increased dementia risk, while nulliparity and having 2 to 4 children showed similar associations compared to primiparous women [36]. In our study, the number of children was similarly associated with dementia risk for women and men. As such, the risk variation in women

appears to be more related to social and behavioural factors involved in parenthood rather than biological factors associated with childbearing. A plausible explanation for this could be related to the additional expenditures and responsibilities associated with the number of dependents, which could lead to economic hardships and increase psychological distress in parents [37]. In particular, mothers are more likely to bear the brunt of childcare than fathers in a low-income household; the impact of parenthood on mothers of lower SES may be more adverse [37].

Our study showed that abortion was associated with a lower risk of dementia, while we did not find any link for stillbirth or miscarriage. The COSMIC study by Jang and colleagues also found the risk of Alzheimer disease in women who had incomplete pregnancies was half that of those who never experienced an incomplete pregnancy [34]; however, incomplete pregnancies in this study encompassed surgical- or medical-induced abortion and spontaneous miscarriage. A Danish register–based cohort study that excluded women who had a surgical and medical abortion found stillbirth was associated with 86% greater risk of dementia, while miscarriage was not associated with dementia [38]. Both studies [34,38] had limitations that precluded the effect of spontaneous miscarriage and abortion from being differentiated. The course of pregnancy and childbirth can have a considerable influence on lifestyle and health [34], although we did not find any effect modifications to explain some of the findings in pregnancy-related factors. Further clarification for the mechanism which underpins these observations is needed.

## Exogenous hormone use

The link between premenopausal hormone use and dementia risk has hardly been characterised [39]. A previous study suggested that women who reported hormonal contraceptive use performed better in the visuospatial ability and speed and flexibility domains of the neuropsychiatric tests than those who had never used hormonal contraception [39]. On the other hand, the potential benefits of HRT to prolong estrogen supply in older women have not been corroborated by interventional studies [40,41], while the observational evidence remains conflicting [4,42–45]. The Women's Health Initiative Study (WHIS), the only clinical trial that evaluated postmenopausal hormone therapy on preventing dementia, concluded that the risk of dementia was doubled in women randomised to estrogen-progestin based HRT [40]. In a case–control study in Finland, long-term use of systemic HRT was associated with an increased risk of Alzheimer disease [45]. It is still largely contentious whether HRT can potentially prevent or increase the risk of dementia. The timing of HRT use may be crucial, such that there may be a critical window which exogenous hormone use can confer cognitive benefits in postmenopausal women [35,46]. Our study findings on HRT do not support associations between HRT and dementia risk, nor the aforementioned "timing hypothesis" for HRT initiation in relation to menopause for dementia risk. Consistently, a recent nested case–control study using a UK general practice cohort also found no evidence of an increased risk of dementia associated with menopausal hormone therapy; no evidence of different time of hormone therapy initiation may pose different risk of dementia [47].

## Strengths and limitations

The strengths of our study were the large sample size, with linkage to national health records and death registers. Further, our study included a comprehensive list of reproductive factors and exogenous hormone use through the life course. The limitations included retrospective and self-reported measures of reproductive factors, which may be inherently subject to

recall bias and misclassification. Socioeconomic position has been associated with several reproductive factors, such as age at first birth and parity [48]. After adjusting for Townsend deprivation index and weighting according to social structure in our study, the associations between reproductive factors and the risk of dementia attenuated. The limited representativeness of the UK Biobank population, which is a cohort predominantly of relatively healthy and affluent people of white ethnic background, means that it is unlikely to produce reliable estimates of either the prevalence of female reproductive factors or the risk of dementia in the UK population at large. Our finding of an interaction between social deprivation and certain reproductive factors suggests that the main results for these factors need to be treated with caution, although any error in estimated effect size appears to be minor. Further, the adjudication of incident dementia in the UK Biobank used the date recorded in the death registry and hospital admission as the incidence date. Age and SES, among other factors, might influence whether dementia is being recorded. It is also possible that some dementia cases were not picked up from hospital admission and death registry data, resulting in underreporting of dementia cases. Dementia subtypes were not differentiated due to the currently small number of events in the UK Biobank. Last, although multiple adjustments were made to account for confounders, there may still be other unmeasured factors that can lead to residual confounding.

## Implications and next steps for research, clinical practice, and public policy

Our research supports a life course approach for dementia prevention, particularly around the design of risk reduction strategies pertaining to reproductive factors which are unique to women. It is necessary to validate our findings on exogenous hormone use through rigorous clinical trials, and our findings may be helpful for identifying high-risk women to participate in future trials. Further, social deprivation is likely to be an important determinant of dementia risk and other aspects of women's health, given that the elevated dementia risk associated with early (natural and artificial) menopause were more strongly associated with dementia in women of lower SES.

## Conclusions

This study highlights that the reproductive and endocrine milieu in women may be involved in dementia risk, although the physical experience of childbearing is unlikely to account for the risk variation in women, given the similar associations observed for number of children and dementia in women and men.

## Supporting information

**S1 Checklist. STROBE statement—Checklist of items that should be included in reports of observational studies.** STROBE, Strengthening the Reporting of Observational Studies in Epidemiology.
(DOCX)

**S1 Table. Number of missingness for each reproductive factor of interest in the UK Biobank.** [a]Collected in women who indicated that they ever had given birth to more than 1 child ($N$ = 184,876). [b]Collected in women who indicated that they ever had been pregnant ($N$ = 231,352). [c]Collected in women who indicated that their periods had stopped (had natural menopause) ($N$ = 165,301). [d]Collected in women who indicated that they had taken the contraceptive pill ($N$ = 220,344). [e]Collected in women who indicated that they had ever used HRT

($N$ = 104,133). [f]Collected in women who indicated that they had ever used HRT and not currently using HRT ($N$ = 87,413). HRT, hormone replacement therapy.
(DOCX)

**S2 Table. Unadjusted and multiple adjusted HRs and women-to-men ratio of HRs (95% CIs) for the risk of dementia associated with number of children for women and men.**
[a]Analyses were adjusted for age, Townsend index, ethnicity, smoking status, systolic blood pressure, BMI, diabetes, total cholesterol, antihypertensive drugs, and lipid-lowering drug. BMI, body mass index; CI, confidence interval; HR, hazard ratio; RHR, ratio of hazard ratio.
(DOCX)

**S3 Table. Unadjusted HRs (95% CIs) for the risk of dementia associated with reproductive factors, by age, SES, smoking, and BMI.** [a]Early menarche was defined as age at first menstrual period before the age of 12 years. [b]Each live birth in women and each child fathered in men. [c]Early menopause was defined as the permanent absence of menstrual periods before the age of 47 years. BMI, body mass index; CI, confidence interval; HR, hazard ratio; HRT, hormone replacement therapy; SES, socioeconomic status.
(DOCX)

**S4 Table. Unadjusted and multiple adjusted HRs for the risk of dementia associated with history and timing of hysterectomy and oophorectomy.** [a]Analyses were adjusted for age, Townsend index, ethnicity, smoking status, systolic blood pressure, BMI, diabetes, total cholesterol, antihypertensive drugs, and lipid-lowering drugs. BMI, body mass index; CI, confidence interval; HR, hazard ratio; SES, socioeconomic status.
(DOCX)

**S5 Table. Unadjusted and multiple adjusted HRs (95% CIs) for the risk of dementia associated with the timing of HRT use in relation to age at (natural or artificial) menopause in postmenopausal women.** [a]Analyses were adjusted for age, Townsend index, ethnicity, smoking status, systolic blood pressure, BMI, diabetes, total cholesterol, antihypertensive drugs, and lipid-lowering drugs. BMI, body mass index; CI, confidence interval; HR, hazard ratio; HRT, hormone replacement therapy.
(DOCX)

**S6 Table. Unadjusted and multiple adjusted HRs (95% CIs) for the risk of dementia associated with age at menarche, reproductive years and age at menopause, after excluding those had hysterectomy or oophorectomy.** [a]Analyses were adjusted for age, Townsend index, ethnicity, smoking status, systolic blood pressure, BMI, diabetes, total cholesterol, antihypertensive drugs, and lipid-lowering drugs. BMI, body mass index; CI, confidence interval; HR, hazard ratio.
(DOCX)

**S7 Table. Multiple adjusted risk ratios (95% CIs) for the risk of dementia associated with reproductive factors, comparing Cox proportional hazards regression (HRs), competing risk model (HRs), and multinomial regression (odds ratios).** Analyses were adjusted for age, Townsend index, ethnicity, smoking status, systolic blood pressure, BMI, diabetes, total cholesterol, antihypertensive drugs, and lipid-lowering drugs. [a]These CIs were calculated without the floating absolute risk, to provide comparable results to the competing risk models and the multinomial regression models. BMI, body mass index; CI, confidence interval; HR, hazard ratio; HRT, hormone replacement therapy; OR, odds ratio.
(DOCX)

**S8 Table. Multiple adjusted HRs (95% CIs) for the risk of dementia associated with reproductive factors stratified by Townsend fifths.** Analyses were adjusted for age, Townsend index, ethnicity, smoking status, systolic blood pressure, BMI, diabetes, total cholesterol, antihypertensive drugs, and lipid-lowering drugs. [a]Early menarche was defined as age at first menstrual period before the age of 12 years. [b]Each live birth in women and each child fathered in men. [c]Early menopause was defined as the permanent absence of menstrual periods before the age of 47 years. BMI, body mass index; CI, confidence interval; HRT, hormone replacement therapy.
(DOCX)

**S9 Table. Townsend-weighted multiple adjusted HRs (95% CIs) for the risk of dementia associated with reproductive factors.** Analyses were adjusted for age, Townsend index, ethnicity, smoking status, systolic blood pressure, BMI, diabetes, total cholesterol, antihypertensive drugs, and lipid-lowering drugs. [a]Weighted HR are weighted according to equal weights of each stratified HR, to represent that the national Townsend fifths contain equal number of people. BMI, body mass index; CI, confidence interval; HR, hazard ratio; HRT, hormone replacement therapy.
(DOCX)

**S10 Table. Multiple adjusted HRs (95% CIs) for comparing individual and combined associations between reproductive factors and dementia. (A)** A consideration of pregnancy related factors: number of live births, stillbirths, miscarriages, and abortions, although there were no significant change in the model coefficients in this model compared with the individual associations. Analyses were adjusted for age, Townsend index, ethnicity, smoking status, systolic blood pressure, BMI, diabetes, total cholesterol, antihypertensive drugs, and lipid-lowering drugs. **(B)** A consideration of factors throughout the life span including age at menarche, parous versus not, hysterectomy and or oophorectomy, HRT use, and contraceptive pill use. Analyses were adjusted for age, Townsend index, ethnicity, smoking status, systolic blood pressure, BMI, diabetes, total cholesterol, antihypertensive drugs, and lipid-lowering drugs. BMI, body mass index; CI, confidence interval; HRT, hormone replacement therapy.
(DOCX)

**S11 Table. Multiple adjusted HRs for the risk of dementia associated with reproductive factors in women, after imputed for missing data using MICE, compared with complete case analyses.** Analyses were adjusted for age, Townsend index, ethnicity, smoking status, systolic blood pressure, BMI, diabetes, total cholesterol, antihypertensive drugs, and lipid-lowering drugs. BMI, body mass index; CI, confidence interval; HR, hazard ratio; HRT, hormone replacement therapy; MICE, Multivariate Imputation by Chained Equations.
(DOCX)

## Author Contributions

**Conceptualization:** Jessica Gong, Sanne A. E. Peters, Mark Woodward.

**Data curation:** Mark Woodward.

**Formal analysis:** Jessica Gong, Katie Harris, Sanne A. E. Peters.

**Investigation:** Jessica Gong, Katie Harris, Sanne A. E. Peters, Mark Woodward.

**Methodology:** Mark Woodward.

**Software:** Jessica Gong, Katie Harris.

**Supervision:** Katie Harris, Sanne A. E. Peters, Mark Woodward.

**Visualization:** Jessica Gong, Katie Harris.

**Writing – original draft:** Jessica Gong.

**Writing – review & editing:** Katie Harris, Sanne A. E. Peters, Mark Woodward.

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
