## [Editor Report · Decision Letter 0]

13 Apr 2021

Dear Dr Gong, 

Thank you for submitting your manuscript entitled "Reproductive factors and the risk of incident dementia: Results from the UK Biobank" for consideration by PLOS Medicine.

Your manuscript has now been evaluated by the PLOS Medicine editorial staff and I am writing to let you know that we would like to send your submission out for external peer review.

Please re-submit your manuscript within two working days, i.e. by Apr 15 2021 11:59PM.

Kind regards,

Caitlin Moyer, Ph.D.

Associate Editor

PLOS Medicine

---

## [Decision Letter · Decision Letter 1]

13 Oct 2021

Dear Dr. Gong,

Thank you very much for submitting your manuscript "Reproductive factors and the risk of incident dementia: Results from the UK Biobank" (PMEDICINE-D-21-01592R1) for consideration at PLOS Medicine. 

Your paper was evaluated by a senior editor and discussed among all the editors here. It was also discussed with an academic editor with relevant expertise, and sent to four independent reviewers, including a statistical reviewer. The reviews are appended at the bottom of this email and any accompanying reviewer attachments can be seen via the link below:

[LINK]

In light of these reviews, I am afraid that we will not be able to accept the manuscript for publication in the journal in its current form, but we will consider a revised version that addresses the reviewers' and editors' comments. Obviously we cannot make any decision about publication until we have seen the revised manuscript and your response, and we plan to seek re-review by one or more of the reviewers. 

We expect to receive your revised manuscript by Nov 03 2021 11:59PM. Please email us (plosmedicine@plos.org) if you have any questions or concerns.

We look forward to receiving your revised manuscript. 

Sincerely,

Caitlin Moyer, Ph.D.

Associate Editor

PLOS Medicine

plosmedicine.org

1. Please address the following comment from the Academic Editor: The paper gives a detailed look at this issue, but there are a couple of major areas of concern – one is the challenge of interpretation of the results given the low response rate for UK Biobank and the likelihood that those who took part from lower SES seem unlikely to fully represent these populations. The findings suggest residual confounding with these factors.

2. Title: Please revise your title according to PLOS Medicine's style. Your title must be nondeclarative and not a question. It should begin with main concept if possible. "Effect of" should be used only if causality can be inferred, i.e., for an RCT. Please place the study design ("A randomized controlled trial," "A retrospective study," "A modelling study," etc.) in the subtitle (ie, after a colon).

3. Abstract Background: Provide the context of why the study is important. The final sentence should clearly state the study question.

4. Abstract: Methods and Findings: Please quantify the main results presented in the abstract with both 95% CIs and p values.

5. Abstract: Methods and Findings: Please mention the important dependent variables that are adjusted for in the analyses.

6. Abstract: Methods and Findings: In the last sentence of the Abstract Methods and Findings section, please describe the main limitation(s) of the study's methodology.

7. Abstract: Conclusions: Please address the study implications without overreaching what can be concluded from the data; the phrase "In this study, we observed ..." may be useful.

8. Author Summary: At this stage, we ask that you include a short, non-technical Author Summary of your research to make findings accessible to a wide audience that includes both scientists and non-scientists. The Author Summary should immediately follow the Abstract in your revised manuscript. This text is subject to editorial change and should be distinct from the scientific abstract. Please see our author guidelines for more information: https://journals.plos.org/plosmedicine/s/revising-your-manuscript#loc-author-summary

9. Main text: For in-text citations, please note reference numbers within square brackets, placed before the sentence punctuation, rather than superscript numbers. Where multiple references are listed, please do not include spaces within brackets.

10. Introduction: Line 63: Please expand the description of the study’s objective to include that men were also included in the investigation.

11. Methods: Please ensure that the study is reported according to the STROBE guideline. Please add the following statement, or similar, to the Methods: "This study is reported as per the Strengthening the Reporting of Observational Studies in Epidemiology (STROBE) guideline (S1 Checklist)."

12. Methods: Did your study have a prospective protocol or analysis plan? Please state this (either way) early in the Methods section.

13. Methods: Line 101: Please provide more information on the Townsend Deprivation Index for determining SES and how this was categorized.

14. Results: For all the main results presented in the text, please note both 95% CIs and p values.

15. Discussion: Please present and organize the Discussion as follows: a short, clear summary of the article's findings; what the study adds to existing research and where and why the results may differ from previous research; strengths and limitations of the study; implications and next steps for research, clinical practice, and/or public policy; one-paragraph conclusion.

16. Discussion: Line 263: Please check if this should read “dendritic spine”.

17. Conclusions: Line 350: Here and throughout, please avoid language that implies causality, such as “This study highlights that the reproductive and endocrine milieu in women can have a significant impact on their dementia risk…” as this study is observational and therefore causality cannot be inferred.

18. Tables and Supporting information tables: For all analyses presented, please include both adjusted and unadjusted analyses. Please report p values as well as 95% CIs.

19. Table 1: Please note all abbreviations used in the legend (SES, HRT, etc).

20. Table 3: Please also present the unadjusted analyses. Please present p values in addition to 95% CIs.

21. Table 4: Please also present the unadjusted analyses. Please define all abbreviations in the legend (HRT, BMI).

22. S1 Table: The rates of dementia by number of children for men could be moved to the main text (the adjusted and unadjusted rates for women are presented in Table 2).

23. Figure 1: Please describe each panel (A, B, C, etc) in the legend. Please also note in the legend the meaning of the line/shaded region.

24. Figure 2: Please note in the legend that the bars represent the 95% CIs.

25. References: Please use the "Vancouver" style for reference formatting (particularly journal title abbreviations), and see our website for other reference guidelines https://journals.plos.org/plosmedicine/s/submission-guidelines#loc-references

26. STROBE Checklist: Thank you for including the checklist. Please refer to locations in the text using section and paragraph numbers, rather than page numbers.

Comments from the reviewers:

Reviewer #1: This paper investigates how reproductive factors are related to incident dementia

The analyses are based on the UK biobank

This a a well written paper but there are several potential methodological issues that are not addressed 

- Firstly as authors are modeling age at dementia they need to make sure that the information used is accurate. In this case they use age at dementia based on the first time dementia is recorded and that includes death

This is very likely to be biased as dementia is not fatal. This definition would need to be challenged in sensitivity analyses

- the authors report higher incidence of dementia in men than in women which is in disagreement with most reports. How do they explain that ?

- The authors use multiple indicators of reproductive life and this poses two problems : first a DAG would be useful to understand the underlying mechanisms that are investigated and this will help the reader to understand the analytical strategy , second the authors might condider correcting for multiple comparisons if they don't explain their strategy better

- the authors would need to take into account death as a competing vent

Reviewer #2: This is a very interesting and important manuscript examining reproductive risk factors in women and men using UK Biobank data. The authors report that several aspects of reproduction in women including age at menarche, age at menopause, hysterectomy, oral contraceptive use, and hormone replacement therapy are associated with risk of dementia. Further, there were U-shaped associations between number of children and risk of dementia for both women and men. Overall, the manuscript is well-written. My comments are enumerated below.

1. A major concern is the lack of consideration of race/ethnicity. It is well known that age of menarche and menopause tend to occur earlier in women who are black as compared to white. Thus, race/ethnicity must be considered in the analyses. It should be reported in the tables and stratification is warranted.

2. The Townsend deprivation index is included as a covariate, but education is not. Although these should be related, it is also important to assess the effect of the number of years of education separately. Women with age at first birth who were aged 24 or less were at greater risk of developing dementia - this could partially be influenced by education. Further, women with an abortion were at reduced risk of dementia - could this also be related to education?

3. Premenopausal bilateral oophorectomy, but not unilateral oophorectomy, has been associated with risk of dementia. Is it possible to determine whether the oophorectomy was unilateral or bilateral?

4. There is no discussion of missingness. Presumably with this large of study, some data was missing. How was this handled? Were individuals excluded (in which case how did they compare to those included), or was missing data imputed?

5. What is meant by incident 'fatal' dementia? I have not heard this term before. Does this mean it was only reported on the death certificate but not the medical record? 

6. Was the duration of contraceptive use associated with risk of dementia?

7. Given HRT and contraceptive use was ascertained for women, could the authors also look at testosterone use for men?

8. Timing of HRT in relation to menopause is important. Is there information regarding when HRT was started in relation to menopause? Several studies of women who start HRT around the time of menopause do not report an increased risk of cognitive decline or dementia.

9. Each of the reproductive factors were examined in isolation but that is not how they occur over the lifespan for women. It would be helpful to show the combination of factors or to put multiple in a model to determine which aspects might be most important. For example, is hysterectomy associated with dementia because it often co-occurs with bilateral oophorectomy and results in an earlier age at menopause?

Reviewer #3: The overall significance of this paper is very high. Female-specific reproductive factors may explain sex differences in dementia prevalence. Further, the effects of oral contraceptives on dementia risk are unknown. Thus, the current study brings exciting new data to the field while appropriately addressing limitations of the study. Only minor revisions are suggested to consider additional factors in data interpretation and expand on a few discussion points.

Main Comments:

1. It is stated several times through the abstract, results, and discussion that for early menopause hysterectomy or oophorectomy that "women of lower SES had a greater dementia risk, but not for women of high SES." This statement could be misleading. Table 4 shows that even in high SES women that early menopause was still associated with increased risk (1.19 HR). Thus, it seems more accurate to convey that early menopause increased dementia risk, but that low SES women had increased risk relative to high SES women. 

2. Please discuss how HRT age of initiation could have played a role (if at all) in the SES difference? Did women with higher SES go into menopause earlier or later? Did they initiate HRT earlier or did a greater percentage of them undergo HRT? Could that have contributed to risk? 

3.In Figure 1 it looks like some women started HRT very early - at 35 years of age and these were the women at highest risk (2.0 HR). Did these women also have premature menopause (either natural or surgical) that could have increased their risk? How was this accounted for in the interpretation of the data? For example, if women went through HRT early due to premature menopause (surgical or natural) then how was the risk due to early menopause vs. HRT separated statistically?

Minor Comments: 

1. In the abstract, the statement "although female biological factors involved in childbearing are unlikely to account for risk variation" was unclear on it is own. It would be helpful to say why. 

3. In the final paragraph it is unclear what is meant by "Further, deprivation is likely to an important determinant of dementia". Deprivation of what? Estrogen?

4. The final sentence it is unclear what the "Beijing Declaration" is. This ending to the manuscript felt odd.

Reviewer #4: See attachment

Michael Dewey

[LINK]

---

## [Decision Letter · Decision Letter 2]

23 Dec 2021

Dear Dr. Gong,

Thank you very much for submitting your manuscript "Reproductive factors and the risk of incident dementia: a prospective cohort study of UK Biobank participants" (PMEDICINE-D-21-01592R2) for consideration at PLOS Medicine. 

Your revised paper was evaluated by a senior editor and discussed among all the editors here. It was also discussed with an academic editor with relevant expertise, and sent to three of the original reviewers, including a statistical reviewer. The reviews are appended at the bottom of this email and any accompanying reviewer attachments can be seen via the link below:

[LINK]

In light of the remaining points mentioned by Reviewer 4, I am afraid that we will not be able to accept the manuscript for publication in the journal in its current form, but we would like to consider a revised version that addresses the reviewers' and editors' comments. Obviously we cannot make any decision about publication until we have seen the revised manuscript and your response, and we plan to seek re-review by one or more of the reviewers. 

We expect to receive your revised manuscript by Jan 13 2022 11:59PM. Please email us (plosmedicine@plos.org) if you have any questions or concerns.

We look forward to receiving your revised manuscript. 

Sincerely,

Caitlin Moyer, Ph.D.

Associate Editor

PLOS Medicine

plosmedicine.org

1. From the Academic Editor: In terms of discussing the limitation of the representativeness of UK Biobank, please describe specific evidence on socioeconomic and other factors that might covary with each of the variables, e.g. abortion, stillbirth, early pregnancy, etc. It seems as if the adjustment for Townsend Deprivation Index and weighting according to social structure has in fact attenuated the statistical significance of associations. Please provide additional comment and interpretation for this result. Please discuss differential responses by deprivation and within different communities that could be related to some of the factors.

Please discuss the limitations of the methods for determining dementia outcomes in terms of bias (including SES and age-related biases) and under-reporting.

2. Response to reviewers: Please completely address the remaining concerns of Reviewer 4, including the points regarding multiple imputation as well as the inconsistencies with the interpretation of observed U shaped relationships.

3. Title: Please capitalize the first word of the subtitle, and please remove “prospective” from the title: “Reproductive factors and the risk of incident dementia: A cohort study of UK Biobank participants”

4. Data Availability statement: Please omit the part of the statement indicating that UK Biobank resources are available from the authors, but do include the most direct link possible for data access requests (e.g. https://www.ukbiobank.ac.uk/enable-your-research/apply-for-access) and also include any relevant (non-author) contacts (e.g. access@ukbiobank.ac.uk).

5. Abstract: Please combine the Methods and Findings sections into one section, “Methods and findings”.

6. Abstract: Line 37: Please change to “1866 dementia cases were recorded in women” if this is intended.

7. Abstract: Please report exact p values, except where p<0.001.

8. Author summary: What do these findings mean? Please format this section into 2-3 bulleted points.

9. Introduction: Line 95: Please avoid use of “effects” as this implies causality.

10. Methods: Line 100: Please mention that there was no prospective analysis plan designed specifically for this study, though the framework for the design was drawn from the analyses carried out in a previous study.

11. Methods: Line 153: Please clarify if this should be “including information about unemployment…” or similar.

12. Results: When reporting p values, please report the exact p value in the text, except please report p<0.001 where applicable. Please provide the exact p value where p<0.01.

13. Results: Line 330-331: Please revise to avoid casual implications “...no evidence of HRT duration affecting dementia risk…” and please avoid causal language throughout.

14. Discussion: Line 381-383: Please clarify this sentence as the meaning is not clear. “The discrepancy in findings cannot be explained by the exclusion of women who reported hysterectomy or oophorectomy in the two studies reported opposite findings”

15. Discussion: Line 401: Please revise to avoid describing causal interpretations of the study findings.

16. Discussion: Line 459: Please remove “...and the prospective design…” from the sentence.

17. Page 32: Please remove the sections Availability of data and materials, Financial disclosure statement, Competing interests, and Author contributions from the main text of the manuscript. Please make sure the information is accurately and completely entered in the relevant fields of the manuscript submission system.

18. Table 3, Table S2: Please report exact p values, unless p<0.001.

19. STROBE Checklist: Thank you for including the checklist. Please revise, using only section and paragraph numbers to refer to locations within the text (e.g. Methods, paragraph 1). Please do not include page or line numbers.

20. S3 Table, S4 Table, S6 Table: Please report p values along with the 95% CIs for these analyses.

21. S5 Table, S2 Figure: Please report exact p values, unless p<0.001.

22. S1 Figure and S2 Figure: Please also include a column with the HR, 95% CIs and p values from the unadjusted models.

23. Reference list: Please double check the reference formatting, and please see our website for additional guidance: https://journals.plos.org/plosmedicine/s/submission-guidelines#loc-references

For example, it seems as if the journal title for reference 7 should be Am J Obstet Gynecol.

Comments from the reviewers:

Reviewer #2: The authors have addressed all of my comments.

Reviewer #3: All comments were addressed. 

Reviewer #4: See attachment

Michael Dewey

[LINK]

---

## [Decision Letter · Decision Letter 3]

10 Feb 2022

Dear Dr. Gong,

Thank you very much for re-submitting your manuscript "Reproductive factors and the risk of incident dementia: A cohort study of UK Biobank participants" (PMEDICINE-D-21-01592R3) for review by PLOS Medicine.

I have discussed the paper with my colleagues and the academic editor and it was also seen again by one of the reviewers. I am pleased to say that provided the remaining editorial and production issues are dealt with we are planning to accept the paper for publication in the journal.

[LINK]

We look forward to receiving the revised manuscript by Feb 17 2022 11:59PM.   

Sincerely,

Caitlin Moyer, Ph.D.

Associate Editor 

PLOS Medicine

plosmedicine.org

Requests from Editors:

1. Throughout (Abstract Line 45-46, Line 52-53; Author Summary Lines 69-70 and 73-74; Results Lines 287-288; Discussion Lines 425-428 , Conclusion Lines 497-499): A similar U-shaped association between number of children and the risk of dementia for both sexes is mentioned as a finding. The evidence in support of an overall relationship between number of children and risk of dementia was not entirely clear from the results. An apparent U-shaped pattern can be observed in Figure 2. The results seem to suggest a significant difference between women with no compared to 2 live births in the adjusted analyses, while there is no significant association with parity, or per live birth (Table 2). It is not completely clear, but from Figure 2, this may not be the case in men, and this would seem to suggest the relationship may differ between sexes to some extent. Also, Table 3 and Table S7 seem to suggest there may be different relationships between number of births and risk when SES/deprivation are examined, for women and men. Please clarify this throughout the manuscript, and revise conclusions as appropriate.

2. Abstract: Line 27: Please change “equivocal” to “uncertain” in this sentence.

3. Abstract: Line 29: Please revise to avoid the word “effects” as this may imply causality. We suggest: “...whether the associations vary by age, socioeconomic status…”

4. Abstract: Line 40: We suggest referring to either >14 or >/= 15 consistently throughout the manuscript. In the results, >14 yrs is used.

5. Abstract: Please combine the “Methods and Findings” and “Results” sections into a single “Methods and Findings” section.

6. Abstract: Line 45-46: For this sentence, please report whether there is a significant association between number of children and dementia risk, provide the results (HR and 95% CIs, p values) to support the mentioned associations between number of children and dementia risk. “The U-shaped associations between the number of children and the risk of dementia were similar for both sexes,”

7. Abstract: Line 46-47: Please mention that this association did not reach statistical significance for early menopause.

8. Abstract: Line 48-50: We suggest also mentioning the representativeness of UK Biobank population as one of the main limitations of the study.

9. Abstract: Line 52-53: “...and there was a similar association between the number of children and dementia risk between women and men.” We suggest removing or clarifying this point to avoid misinterpretation (please see point #1 above).

10. Author summary: Line 59: We suggest “uncertain” rather than “equivocal” in this point.

11. Author summary: Line 69: We suggest revising to: “The U-shaped associations between number of children and dementia appeared similar…”

12. Author summary: Line 72: We suggest revising to “In this study we observed that certain reproductive factors are associated with greater risk of dementia. Future work is needed to understand whether this is related to the fact that these factors may be associated with shorter cumulative exposure to endogenous estradiol.” or similar.

13. Author summary: Line 73-74: Please see point #1 above. We suggest revising to: “Findings suggest that risk variation in women may not be associated with factors associated with childbearing because a similar U-shaped pattern was observed between number of children fathered and dementia risk among men.” or similar.

14. Throughout: In-text citations: Please separate the square brackets from the preceding word with a space.

15. Introduction: Line 91: We suggest “uncertain” rather than “equivocal” in the sentence.

16. Results: Line 248: Please define the values in parentheses at the first point of use in the text, e.g. rate (95% confidence interval).

17. Results: Line 287-288: Please revise the sentence to: “Compared with those who had two children, the associations between the number of children and dementia were similar for women and men and appeared to be U-shaped (Figure 2).” Please describe the HR with 95% CIs and p values for associations overall, and separately for men and women.

18. Discussion: Line 364: We suggest revising to: “...with dementia risk later in life.”

19. Discussion: Line 384-385: We suggest: “...estradiol has been shown to be correlated with neuronal dendritic spine density [22]...” or similar, as it seems the reference does not describe regeneration.

20. Discussion: Line 408-409: We suggest revising to: “...providing further support that early cessation of hormones may be associated with poorer cognitive outcomes.” or similar.

21. Discussion: Line 424-425: Please clarify if this should be: “...while nulliparity and having two to four children showed similar associations compared to primiparous women…”

22. Discussion: Line 425-428: It is not clear from the results whether a statistically significant association between number of children and dementia risk was identified for both men and women. We suggest tempering this sentence accordingly. “In our study, the number of children was similarly associated with

dementia risk for women and men. As such, the risk variation in women appears to be more related to social and behavioural factors involved in parenthood rather than biological factors associated with childbearing.”

23. Discussion: Line 460: We suggest revising to: “...do not support associations between HRT and dementia risk…” or similar.

24. Discussion: Line 474: We suggest adding an additional sentence describing the limited representativeness of the UK Biobank, and a clear description of why this is important to consider in terms of interpretations of your results.

25. References: Please update the reference information for Referenes # 40 and # 46.

26. Figure 2: Please provide the results from both adjusted and unadjusted analyses for men (e.g. similar to the results for women are provided in Table 3).

27. S9 Table: Please provide p values in addition to 95% CIs.

Comments from Reviewers:

Reviewer #4: The authors have now addressed my remaining points.

Michael Dewey

[LINK]

---

## [Editor Report · Decision Letter 4]

23 Feb 2022

Dear Dr Gong, 

On behalf of my colleagues and the Academic Editor, Carol Brayne, I am pleased to inform you that we have agreed to publish your manuscript "Reproductive factors and the risk of incident dementia: A cohort study of UK Biobank participants" (PMEDICINE-D-21-01592R4) in PLOS Medicine.

Please also address the following editorial requests:

-Abstract Line 49-50: Please correct the typo to read: “There was evidence that hysterectomy…”

-Abstract: Line 55-56: Please change to: “In this study, we observed that some reproductive events related to shorter cumulative endogenous estrogen exposure…”

-Discussion: Line 487: Please change “...Caucasian ancestry…” to “white ethnic background” if this is what is intended.

-Reference List: Please update reference 46 with the complete citation information.

PRESS

Sincerely, 

Caitlin Moyer, Ph.D. 

Associate Editor 

PLOS Medicine